# Functional Analysis of Three miRNAs in *Agropyron mongolicum* Keng under Drought Stress

Xuting Zhang [1],[†], Bobo Fan [1],[†], Zhuo Yu [1], Lizhen Nie [2], Yan Zhao [3], Xiaoxia Yu [1], Fengcheng Sun [2], Xuefeng Lei [1] and Yanhong Ma [1],[*]

1   Agricultural College, Inner Mongolia Agricultural University, Hohhot 010000, China; zhangxuting0527@163.com (X.Z.); fanbobo19@126.com (B.F.); yuzhuo58@sina.com (Z.Y.); yuxiaoxia1985@sina.com (X.Y.); nmndnxyky@163.com (X.L.)
2   Inner Mongolia Academy of Agricultural & Animal Husbandry Sciences, Hohhot 010000, China; 13734717240@163.com (L.N.); sfcnmnky@sina.com (F.S.)
3   College of Grassland, Resources and Environment, Inner Mongolia Agricultural University, Hohhot 010000, China; zhaoyannmg@sina.com
*   Correspondence: mayanhong80@126.com; Tel.: +86-151-8472-5923
†   These authors contributed equally to this work.

**Abstract:** *Agropyron mongolicum* Keng, a perennial diploid grass with high drought tolerance, belongs to the genus *Agropyron*, tribe *Triticeae*. It has made tremendous contributions toward reseeding natural pasture and seeding artificial grassland in China, especially in the arid and semi-arid area of northern China. As a wild relative of wheat, *A. mongolicum* is also a valuable resource for the genetic improvement of wheat crops. MicroRNAs are small non-coding RNA molecules ubiquitous in plants, which have been involved in responses to a wide variety of stresses including drought, salinity, chilling temperature. To date, little research has been done on drought-responsive miRNAs in *A. mongolicum*. In this study, two miRNA libraries of *A. mongolicum* under drought and normal conditions were constructed, and drought-responsive miRNAs were screened via Solexa high throughput sequencing and bioinformatic analysis. A total of 114 new miRNAs were identified in *A. mongolicum* including 53 conservative and 61 unconservative miRNAs, and 1393 target genes of 98 miRNAs were predicted. Seventeen miRNAs were found to be differentially expressed under drought stress, seven (amo-miR21, amo-miR62, amo-miR82, amo-miR5, amo-miR77, amo-miR44 and amo-miR17) of which were predicted to target on genes involved in drought tolerance. QRT-PCR analysis confirmed the expression changes of the seven drought related miRNAs in *A. mongolicum*. We then transformed the seven miRNAs into *Arabidopsis thaliana* plants, and three of them (amo-miR21, amo-miR5 and amo-miR62) were genetically stable. The three miRNAs demonstrated the same expression pattern in *A. thaliana* as that in *A. mongolicum* under drought stress. Findings from this study will better our understanding of the molecular mechanism of miRNAs in drought tolerance and promote molecular breeding of forage grass with improved adaption to drought.

**Keywords:** *Agropyron mongolicum* Keng; drought-responsive microRNA; NEXT-generation sequencing; target gene prediction; functional analysis

## 1. Introduction

Climate change and global warming are major challenges of our time and currently drought stress has become the predominant cause for decreased crop yield. It has demonstrated a substantial impact on the ecosystem and agriculture in the affected region and harmed the local economy. *Agropyron* is a genus of Eurasian plants in the grass family [1], which are known to be tolerant to drought,

low temperature and salinity, and resistant to the rusts and other pathogens of wheat (*Triticum aestivum* L.), which can be valuable resources in wheat improvement [2,3]. *Agropyron mongolicum* (Poaceae, $2n = 2x = 14$), a representative member of *Agropyron*, is drought tolerant, and grows vigorously in sandy soil on the edge of Hunshandac desert with only 250 mm annual precipitation in Abaga Banner of Inner Mongolia [4]. It has been playing an important role in reseeding natural pasture and seeding artificial grassland in China, especially in arid and semi-arid areas of northern China [5]. Moreover, its cold resistance, drought resistance, salt tolerance, resistance to diseases and pests, and excellent resistance genes provide a genetic resource for genetic modification of food crops (such as wheat and barley) [6]. In past decades, many studies have focused on the classification, genetic diversity, distant hybridization, and double breeding of *A. mongolicum*. Several genes involved in drought tolerance have been isolated. For example, Lea3 gene encoding the late embryogenesis abundant protein was obtained from *A. mongolicum* using homology-based PCR [7]. *MwAP2/EREBP*, isolated through RT-PCR and RACE techniques, demonstrated its involvement in drought-tolerance in *A. mongolicum* Keng [8]. Recently, Ma and associates have identified and evaluated drought-responsive RNAs through transcriptome analysis in *A. mongolicum* [9]. However, to date to the best of our knowledge, no experimentally or computationally identified drought-responsive miRNAs in *A. mongolicum* have been reported.

MicroRNAs (miRNAs) are single-stranded noncoding RNAs ranging in size of approximately 20–24 nucleotides (nt). Accumulating evidence has shown that miRNAs act as key factors in the regulation of gene expression, development, and responses to various biotic and abiotic stresses in plants [10,11]. Drought stress could induce or reduce the production of miRNAs. Some miRNAs decrease the expression of their target genes through miRNA-mediated gene silencing, while others increase the accumulation of their target genes by binding to the corresponding transcription factors [12,13]. For instance, in *Arabidopsis*, miR168, miR394, miR396, miR167, miR165, miR319, miR159, miR156, miR393, miR171, miR158, and miR169 were shown to be drought-responsive [14]. miR168, one of the most commonly detected stress-inducible miRNAs, exists in various plant species. It has been confirmed to control the homeostasis of *ARGONAUTE1* (*AGO1*) under ABA treatment [15]. In another case, miR169 was down-regulated under drought stress which suppressed the cleavage of its target gene *NUCLEAR FACTOR Y A5 (NFYA5)* in *Arabidopsis* [16]. The conserved *Arabidopsis* miR394 was involved in the regulation of plant response to salt and drought stresses in an abscisic acid (ABA)-dependent manner. Silencing of *LCR* mRNA by miR394 was essential to maintain a certain phenotype favorable for the adaption to the abiotic stresses mentioned above [17]. The target gene of miR159 in *Arabidopsis* was a MYB family transcription factor. ABA induced the up-regulation of miR159, leading to the cleavage of MYB33 and MYB101, and thus reduced plant sensitivity to ABA [18]. Fifty-three miRNAs and 23 key drought response genes were found in the regulatory networks in tomato, most of which were transcription factors, such as ERF, MYB, and NAC [19]. miR2118 and miR858 participated in drought stress response by up-regulating OZF1 gene and certain MYB genes that were involved in the regulation of flavonol biosynthesis in *A. mongolicus* [20]. Some miRNAs mediate drought tolerance by regulating enzyme metabolism. For example, during water shortage, miR398 and miR408 in *Medicago truncatula* were up-regulated. This up-regulation led to the down-regulation of copper superoxide dismutase (CSD2) and established a new water balance [21]. As another example, drought stress up-regulated miR474 in maize and reduced the accumulation of a negative regulator of proline. Subsequently, maize seedlings reached to a better adaption to drought stress [22]. Similarly, down-regulation of miR398 in tobacco under drought stress increased the activity of antioxidants such as SOD enzymes [23].

Genome-wide screening has shown to be an effective way to identify miRNAs. For example, miR156, miR474, miR395, miR319, miR171, miR159, miR394, miR169, miR2118, miR399, miR398, miR403, miR399, miR172, miR167, miR1432, miR827, and miR528 have been detected as drought-responsive miRNAs in *Oryza sativa*, *Glycine max*, *Medicago truncatula*, *Triticum dicoccoides*, and *Zea mays* [23–27]. These findings have demonstrated the important role of miRNAs in plant

adaptation to drought stress. The objective of this study is to identify drought-responsive miRNAs in *A. mongolicum* through genome-wide screening with the aim to elucidate the molecular mechanism of drought tolerance and exploit *A. mongolicum* as a genetic resource for wheat improvement.

## 2. Methods

### 2.1. Ethics Statement

The Seeds of *A. mongolicum* were collected from the Ulanqab grassland in Inner Mongolia Province, China, and kindly provided by Dr. Yan Zhao (College of Grassland, Resources and Environment, Inner Mongolia Agricultural University, Hohhot, China). The experimental procedures were approved by the Ethics Committee for Plant Experiments of the Inner Mongolia Agricultural University of China.

### 2.2. Plant Materials and Drought Treatment

Intact seeds were soaked in water for five hours and then sowed in 20 cm diameter and 30 cm deep clay pots with sterilized field soil. A total of 24 pots with 50 seeds per pot were incubated at 28 °C until germination. The seedlings were then grown in a controlled climate growth chamber at 28 °C with 14 h of day light (illumination intensity of 30000Lx, Boxun BIC-300, Shanghai, China) and irrigated daily with a fixed amount of water. The relative humidity was set up as 75%. When the seedlings reached 15 cm high, they were divided into two groups randomly, the control group and the drought stressed group. The control group was irrigated daily as described previously, and the drought-stressed plants were not watered for 21 days. Each treatment was repeated three times. Leaves were sampled at 3, 6, 9, 12, 15, 18, and 21 d post treatment from both groups. All samples were frozen individually in liquid nitrogen and then stored at −80 °C.

### 2.3. Total RNA Extraction, Small RNA Library Construction, and Sequencing

Total RNA was extracted from the leaves of both the control plants and the drought stressed plants by using TRIzol reagent (Invitrogen, Carlsbad, CA, USA) according to the manufacturer's instructions. RNA was treated with DNase I and then purified with QIAquick PCR purification kit (Qiagen, Germantown, MD, USA). RNA samples from the 21-d drought-treated plants and the control plants were used to generate two small RNA libraries by Beijing Biomarker Co., Ltd. (Bejing, China). Library sequencing was done through Illumina HiSeq 2500 platform (San Diego, CA, USA).

### 2.4. Identification of miRNAs and Prediction of Target Genes

Firstly, clean reads were obtained by removing low quality sequencing fragments from raw reads via Solexa sequencer. Then miRNA annotation for each clean read was retrieved by BLASTN search against GenBank and Rfam databases. The reads of rRNA, tRNA, snRNA, scRNA, and snoRNA were removed. The raw sequencing data were submitted to the NCBI SRA datasets under the accession number PRJNA431719.

All clean reads were classified statistically via length, and the base bias was analyzed according to the base preference of known miRNAs. The reads distribution was checked to meet the principles for miRNA prediction. Then, the remaining clean and unique reads were aligned against the latest miRBase database version 20.0 (http://www.mirbase.org/). When the sequenced fragment and the known miRNAs in miRBase were mismatched less than two bases, the fragment was defined as a conserved miRNA. The non-conserved miRNAs of *A. mongolicum* were predicted with miRDeep2 based on stem-loop hairpin structure, dicer enzyme loci and the free energy of the miRNA precursor [28,29].

The newly predicted miRNAs and the *A. mongolicum* reference transcripts database were used to predict the target genes through the TargetFinder. The miRNA target genes prediction rules were described previously [29,30]. The annotation information of the predicted target genes was obtained by BLAST, and the predicted target genes were compared with the Non-redundant (NR) Swiss-Prot, Gene Ontology (GO), Cluster of Orthologous Groups (COG), GO, COG, and Kyoto Encyclopedia of

Genes and Genomes (KEGG) databases (https://www.genome.jp/kegg/) [31]. Then the target genes of 7 drought-responsive miRNAs were predicted in other plant species (*A. thaliana*, *B. distachyon*, *O. sativa*, *T. aestivum*, *Z. mays*) by using psRNATarget (http://plantgrn.noble.org/psRNATarget/) [32]. The parameters used for prediction were set as follows: maximum expectation score, 3.0; length for complementarity, 20 bp; and range of central mismatch, 9–11 nt.

Phylogenetic trees of the target genes were generated using MEGA7.0 and the neighbor-joining (NJ) method with 1000 bootstrap replications

### 2.5. Differential Expression Analysis of miRNAs

FDR <0.01 and Fold Change ≥2 were set up as the standards to screen the differentially expressed miRNAs.

### 2.6. qRT-PCR Analysis

To validate the high-throughput sequencing results, expression of miRNAs was detected by Stem-loop qRT-PCR with samples harvested 3, 6, 9, 12, 15, 18, and 21 d post drought treatment. The untreated sample was included as a control. Reverse transcription reaction was done on a 7300 Real-Time PCR System (Bio-Rad, Hercules, CA). Total RNA of both the treated and the control plants was used to initiate the reaction via miRcute miRNA cDNA kit (TIANGEN KR211, Beijing, China). The reverse transcriptional system included: 10 μL 2 × buffer, 2 μL RNA template, 1.2 μL RT primer (miRNA unified reverse prime sequence was TGGTGTCGTGGAGTCG), 0.2 μL MMLV (200 U/μL), 6.6 μL RNase-Free ddH$_2$O. PCR amplification procedure was set up as follows: 26 °C 30 min, 42 °C 30 min, 85 °C 10 min. All cDNA products were stored at −20 °C

Expression of seven miRNAs was analyzed by qRT-PCR using a published method with some modifications [33]. Specific stem-loop RT primers were designed using Primer premier 5.0 for amo-miR21, amo-miR82, amo-miR62, amo-miR44, amo-miR5, amo-miR17, and amo-miR77 (Table 1). The qRT-PCR was performed on a LightCycler®480 PCR detection system (Roche, German) by using the MiRcute Enhanced microRNAs Fluorescence Quantitative Detection Kit (SYBR Green, Tiangen, Beijing, China). Three biological replicates were done for each treatment and control, and all reactions were run in triplicate. U6 small nuclear RNA (upstream sequence: GGACATCCGATAAAATTGGAACGATA CAG; downstream sequence: AATTTGGACC ATTTCTCGATTTATGCGTGT) was used as the internal control for qRT-PCR. qRT-PCR amplification system included: 10 μL 2 × miRcute Plus miRNA Premix, 0.08 μL mRNA forward primers (20 μmol/L), 0.08 μL mRNA downstream primers (20 μmol/L), 2 μL cDNA template, 0.4 μL Taq DNA polymerase (2.5 U/μL), and 7.44 μL ddH$_2$O. PCR amplification procedure: 95 °C 3 min, 35 cycles of 95 °C 30 s, 60 °C 30 s, and 72 °C 30 s. The relative expression levels of miRNAs were quantified using the $2^{-\Delta\Delta Ct}$ method. Standard deviations were calculated from three biological replicates.

**Table 1.** Primer sequences for qRT-PCR.

| miRNA | Primer Sequence (5′—3′) |
| --- | --- |
| amo-miR21 | Forward: ACACTCCAGCTGGG GAGTGTATGCCCGTATATAT<br>stem-loop RT primer: CTCAACTGGTGTCGTGGAGTCGGCAATTCAGTTGAG ATAGTTT |
| amo-miR82 | Forward: ACACTCCAGCTGGG ATGCTCGCTCCTCTTTCTGT<br>stem-loop RT primer: CTCAACTGGTGTCGTGGAGTCGGCAATTCAGTTGAG GCTCATA |
| amo-miR62 | Forward: ACACTCCAGCTGGG CACGTGCTCGATGAAAT<br>stem-loop RT primer: CTCAACTGGTGTCGTGGAGTCGGCAATTCAGTTGAG AGTCATT |
| amo-miR44 | Forward: ACACTCCAGCTGGG AATGAGGATGATAACA<br>stem-loop RT primer: CTCAACTGGTGTCGTGGAGTCGGCAATTCAGTTGAG GTCTTGT |
| amo-miR5 | Forward: ACACTCCAGCTGGG GGCGGATGTAGC<br>stem-loop RT primer: CTCAACTGGTGTCGTGGAGTCGGCAATTCAGTTGAG TCCACTT |
| amo-miR17 | Forward: ACACTCCAGCTGGG CCCAACGGGCGGTG<br>stem-loop RT primer: CTCAACTGGTGTCGTGGAGTCGGCAATTCAGTTGAGGCCCCAC |
| amo-miR77 | Forward: ACACTCCAGCTGGG CTGCTCCAGCTGCTCA<br>stem-loop RT primer: CTCAACTGGTGTCGTGGAGTCGGCAATTCAGTTGAGCACATGA |

## 2.7. Construction of Expression Vector of Drought-Responsive miRNAs

Total RNA from the drought treated plants for 15 d and the control plants was used to initiate the reverse transcription reaction via FastQuant cDNA First strand synthetic kit (TIANGEN KR106, Beijing, China). Genomic DNA (gDNA) was removed from the total RNA. The gDNA removal system included: 2 μL 5×gDNA Buffer, 2 μL total RNA template, 6 μL RNase-Free ddH$_2$O. Reverse transcriptional PCR system included: 2 μL 10 × Fast RT Buffer, 1 μL RT Enzyme Mix, 2 μL FQ-RT Primer Mix primers, 5 μL RNase-Free ddH$_2$O. All reverse transcriptional PCR products were stored at −20 °C.

*Bam*H I and *Sac* I restriction sites and protection bases were added to the primer sequences when designing the primers using Primer premier 5.0. All primers used for expression vector construction were listed in Table 2. The PCR amplification system included: 1 μL cDNA, 1 μL forward primer and 1 μL reverse primer (10 μmol/L), 10 μL 2 × Taq PCR Master Mix, 7 μL ddH$_2$O. PCR amplification procedure was set up as follows: 94 °C 5 min, 35 cycles of 94 °C 30 s, 59 °C 30 s, 72 °C 30 s, 72 °C 5 min. All PCR products were stored at 4 °C.

**Table 2.** Primer sequences for RT-PCR.

| Primer | Primer Sequences (5′—3′) |
| --- | --- |
| amo-miR21-F | **CG**ggatccACACTCCAGCTGGG GAGTGTATGCCCGTATATAT |
| amo-miR21-RE | **C**gagctcCTCAACTGGTGTCGTGGAGTCGGCAATTCAGTTGAGATAGTTT |
| amo-miR82-F | **CG**ggatccACACTCCAGCTGGG ATGCTCGCTCCTCTTTCTGT |
| amo-miR82-RE | **C**gagctcCTCAACTGGTGTCGTGGAGTCGGCAATTCAGTTGAGGCTCATA |
| amo-miR17-F | **CG**ggatccACACTCCAGCTGGG CCCAACGGGCGGTG |
| amo-miR17-RE | **C**gagctcCTCAACTGGTGTCGTGGAGTCGGCAATTCAGTTGAGGCCCCAC |
| amo-miR5-F | **CG**ggatccACACTCCAGCTGGG GGCGGATGTAGC |
| amo-miR5-RE | **C**gagctcCTCAACTGGTGTCGTGGAGTCGGCAATTCAGTTGAGTCCACTT |
| amo-miR44-F | **CG**ggatccACACTCCAGCTGGG AATGAGGATGATAACA |
| amo-miR44-RE | **C**gagctcCTCAACTGGTGTCGTGGAGTCGGCAATTCAGTTGAGGTCTTGT |
| amo-miR77-F | **CG**ggatccACACTCCAGCTGGG CTGCTCCAGCTGCTCA |
| amo-miR77-RE | **C**gagctcCTCAACTGGTGTCGTGGAGTCGGCAATTCAGTTGAGCACATGA |
| amo-miR62-F | **CG**ggatccACACTCCAGCTGGG CACGTGCTCGATGAAAT |
| amo-miR62-RE | **C**gagctcCTCAACTGGTGTCGTGGAGTCGGCAATTCAGTTGAGAGTCATT |

Note: The lower case indicates the restriction sites and bold letters indicate the protection bases.

*Gus* gene was removed from pBI121 vectors by *Bam*H I and *Sac* I digestion, and pre-miRNA of amo-miRNAs were combined with pBI121 vector. 20 μL pBI121 enzyme digestion system included: 10 μL pBI121 vector, 1 μL *Bam*H I and 1 μL *Sac* I, 2 μL 10 × FastDigest Buffer, 6 μL ddH$_2$O. 30 μL pre-amo-miRNAs enzyme digestion system included: 20 μL PCR products, 1 μL *Bam*H I and 1 μL

*Sac* I, 3 μL 10 × FastDigest Buffer, 5 μL ddH$_2$O. All digestion systems were incubated at 37 °C for 2 h. The enzyme digestion products were collected by 2% agarose gel electrophoresis. The ligation system included: 3 μL vector fragment, 9 μL insertion fragment, 0.5 μL T4 DNA Ligase, 2 μL 10 × T4 DNA Ligation Buffer, 5.5 μL ddH$_2$O. The ligation reaction was incubated at 16 °C for 24 h. Ligated products were transformed into competent *Escherichia coli* DH5α cells with heat shock method. Positive transformants were selected on LB agar plates containing Kan.

### 2.8. Agrobacterium-Mediated Genetic Transformation of Arabidopsis thaliana

*Agrobacterium* strain LBA4404 was streaked on a LB agar plate containing Rif and Str and incubated at 28 °C for 48 h. A single colony was picked up and inoculated to LB broth. When the OD600 value of the culture reached to 0.5, *Agrobacterium* competent cells were prepared as described previously [32]. Expression vector containing the insert (10 μL) was added to the competent cells and incubated in an ice bath for 30 min. *Agrobacterium* transformation was performed as described previously [34,35].

After the positively transformed *Agrobacteria* were identified, *Arabidopsis* plants were transformed using the floral dip method [36]. T$_0$ generation seeds were harvested from the positively transformed plants. Next, stably transformed *Arabidopsis* plants were screened to T$_3$ generation according to Mendel's law. Total RNAs were extracted from the leaves of three biological replicates of the transgenic plants at flowering stage by using Trizol reagent (Invitrogen, Carlsbad, CA, USA), and then the miRNA in transgenic *A. thaliana* were detected with RT-PCR.

### 2.9. Drought Treatment of Transgenic Arabidopsis Plants

Transgenic *Arabidopsis* plants were grown in 10 cm diameter and 6.5 cm deep clay pots. 100 mL 15% PEG 6000 solution was added in each pot for drought treatment at the flowering period. Then the leaves of transgenic *A. thaliana* were harvested at 0, 3, 6, 9, 18, 21, and 24 h post drought treatment. All samples were frozen individually in liquid nitrogen before stored at −80 °C. The total RNA was extracted by Trizol reagent (Invitrogen, Carlsbad, CA, USA), and was used to initiate the reverse transcription reaction with the FastQuant cDNA First strand synthetic kit (TIANGEN KR106, Beijing, China). The cDNA was then used as the template to detect the expression of drought-responsive miRNAs.

## 3. Results

### 3.1. Sequencing and Identification of A. mongolicum Small RNAs

To investigate the role of *A. mongolicum* miRNAs (amo-miRNAs) in responding to drought stress, the control plants and the drought-stressed plants were used to generate two small RNA libraries, and then the libraries were sequenced via Solexa high-throughput sequencing. The numbers of raw reads from the control plants and the drought-stressed plants were 16.89 M and 30.26 M respectively (Table 3). After removing the low-quality reads, the reads with 'N', and the sequences shorter than 18 nt, the control plants and drought-stressed plants generated a total of 8.05 M and 16.41 M clean reads and accounted for 47.64% and 54.22% of total reads respectively.

**Table 3.** Analysis of deep sequencing data of miRNAs in *A. Mongolicum.*

| Type | Control Plants | | Drought-Stressed Plants | |
|---|---|---|---|---|
| | Reads Number | Percentage | Reads Number | Percentage |
| Total reads number | 16,887,273 | 100% | 30,258,338 | 100% |
| Filter low quality reads | 0 | 0% | 0 | 0% |
| Filter having 'N' reads | 1138 | 0.01% | 3408 | 0.01% |
| Length <18 | 6,681,521 | 39.57% | 10,989,611 | 36.32% |
| Length >30 | 2,159,377 | 12.79% | 2,858,115 | 9.45% |
| Clean Reads | 8,045,237 | 47.64% | 16,407,204 | 54.22% |

The sequences ranging from 18 to 30 nt in *A. mongolicum* were further analyzed. The length distribution of the two libraries were similar overall, and 21 nt and 24 nt displayed the highest redundancy (Figure 1).

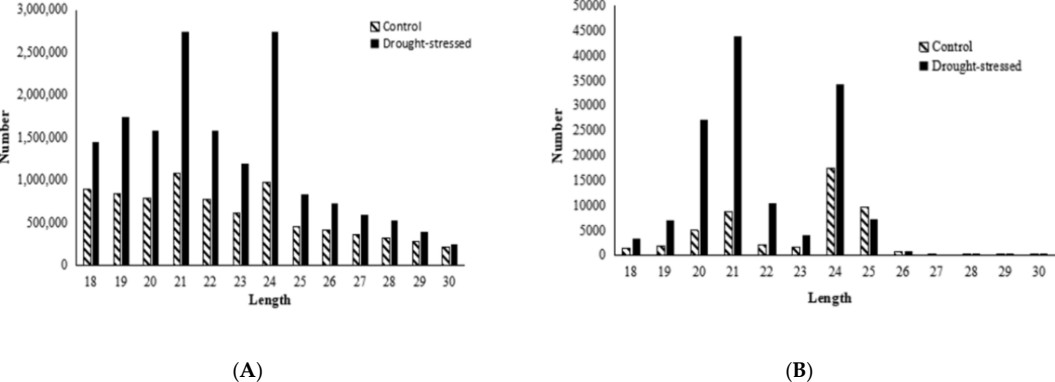

(**A**)                  (**B**)

**Figure 1.** Size distribution of Clean reads (**A**) and miRNA (**B**) of control and drought stress in *A. mongolicum.*

### 3.2. Identification of miRNAs in A. mongolicum

Since the whole genome sequence of *A. mongolicum* is not available, we have used the wheat EST as a reference to search novel miRNAs in the two miRNA libraries. A total of 114 novel miRNAs were identified. The length of the miRNAs ranged from 18 to 25 nt. In order to understand the conservation of the newly identified miRNAs, we compared the novel miRNAs to the genomes of several important plant species, including maize, *Arabidopsis*, rice, *Brachypodium distachyon* and soybean. The results revealed that 61 of the 114 miRNAs were *A. mongolicum* specific and the rest were conserved cross species (Tables S1 and S2).

### 3.3. Differential Expression of miRNAs under Drought Stress

To identify differentially expressed miRNAs, the expression levels of conserved and novel miRNAs under drought treatment were compared with those under normal conditions. 17 conserved and novel miRNAs were detected to be differentially expressed including ten up-regulated and seven down-regulated ones (Table 4). The length of the 17 miRNAs ranged from 18 to 24 nt. Overall, the precursors of these miRNAs had low values of minimal free energy (MFE). Amo-miR63 had the highest MFE of −32.50 kcal/mol., and most of the precursors had minimal free energy index (MFEI) around 0.8. All precursors of the miRNAs were able to form stem loop structures (Figure 2), which was consistent with the structural features of miRNAs. Among the 17 differentially expressed miRNAs, six were conserved in other plant species. Amo-miR21, amo-miR40, amo-miR77, and amo-miR82 shared high homology with *Brachypodium distachyon* miRNA; amo-miR44 was homologous with *A. thaliana* ath-miR854a; and amo-miR62 was homologous with zma-miR164g-3p in corn. The remaining 11 miRNAs were *A. mongolicum* specific (Table 4). Clustering analysis of the 17 differentially expressed miRNAs was shown in Figure 3. Seven miRNAs (amo-miR14, amo-miR17, amo-miR18, amo-miR35, amo-miR40, amo-miR43, and amo-miR44) were found to be down-regulated and ten miRNAs were found to be up-regulated in *A. mongolicum* under drought stress compared to the control.

Table 4. Summary of 17 differentially expressed miRNAs in *A. Mongolicum*.

| miRNA | Sequence | Length (nt) | Regulation Mode | Precursor Length (nt) | MFE (kcal/mol) | MFEI | Conservatism | Homologous miRNA | Expression |
|---|---|---|---|---|---|---|---|---|---|
| amo-miR5 | ggcggauguagccaagugga | 20 | up | 109 | −48.84 | 0.86 | unconservative | | qRT-PCR |
| amo-miR14 | cgcggcggcgggggcggug | 19 | down | 108 | −63.70 | 0.76 | unconservative | | |
| amo-miR17 | cccaacgggcggugggggc | 18 | down | 111 | −66.15 | 0.81 | unconservative | | qRT-PCR |
| amo-miR18 | gauggcgcgcaggcggacg | 19 | down | 108 | −62.43 | 0.72 | unconservative | | |
| amo-miR21 | aaagugucguagaaaaaacuau | 22 | up | 111 | −32.73 | 1.09 | conservative | bdi-miR5180b | qRT-PCR |
| amo-miR29 | gacugaugucgguauggaaccagu | 24 | up | 110 | −41.70 | 0.76 | unconservative | | |
| amo-miR32 | cuacgcgucggaugcacugcgu | 22 | up | 111 | −60.18 | 0.85 | unconservative | | |
| amo-miR35 | aucaaggaauuugugagg | 18 | down | 108 | −59.86 | 1.33 | unconservative | | |
| amo-miR40 | uuugaacuggugguugaaugc | 21 | down | 110 | −42.76 | 0.82 | conservative | bdi-miR7743-3p | |
| amo-miR43 | cgcggcgacgggggcgug | 18 | down | 108 | −61.76 | 0.74 | unconservative | | |
| amo-miR44 | aaugaggaugauaacaagac | 20 | down | 110 | −34.09 | 0.62 | conservative | ath-miR854a | qRT-PCR |
| amo-miR46 | auccgucgugauaugaaaaccagc | 24 | up | 114 | −61.36 | 0.93 | unconservative | | |
| amo-miR49 | cgccggagcugcaaugaagc | 20 | up | 109 | −62.25 | 0.93 | unconservative | | |
| amo-miR62 | cacgugcucgaugaaaugacu | 21 | up | 110 | −46.04 | 0.92 | conservative | zma-miR164g-3p | qRT-PCR |
| amo-miR63 | uugcgucaaagguccuagau | 20 | up | 110 | −32.50 | 0.93 | unconservative | | |
| amo-miR77 | cugcuccagcugcucaugug | 20 | up | 108 | −66.79 | 1.04 | conservative | bdi-miR7725b-5p.2 | qRT-PCR |
| amo-miR82 | gaguguaugcccguauauaugagc | 24 | up | 114 | −33.62 | 0.69 | conservative | bdi-miR5066 | qRT-PCR |

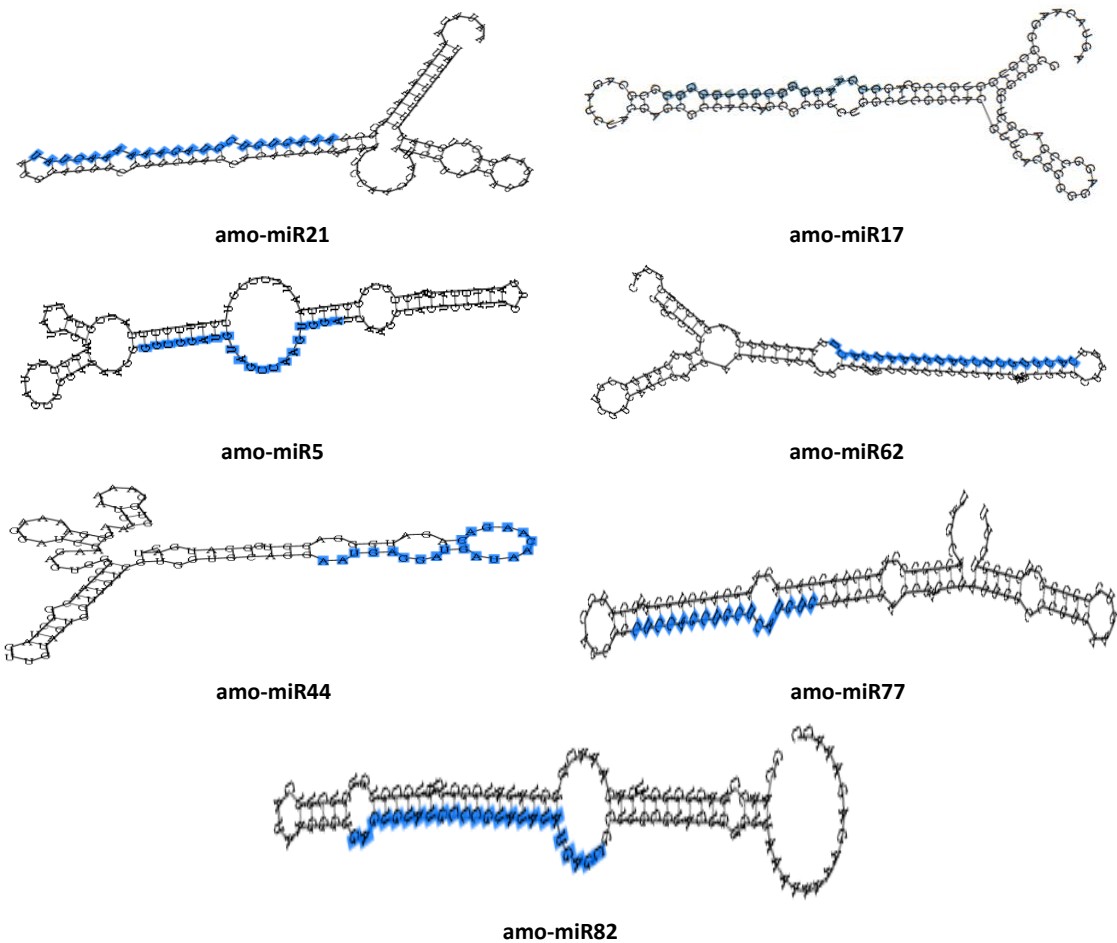

**Figure 2.** Secondary structure of seven novel miRNAs identified in *A. Mongolicum*.

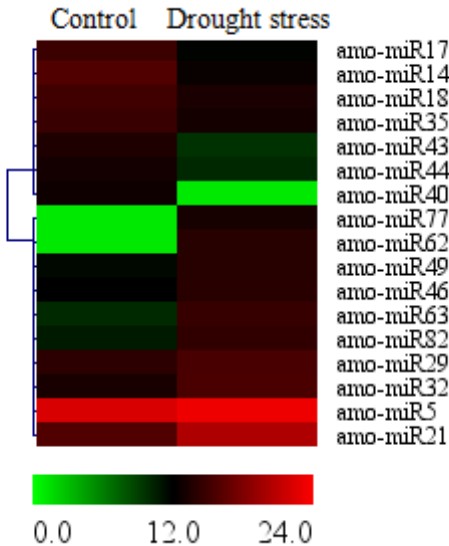

**Figure 3.** Expression profile of 17 miRNAs under drought and normal conditions.

*3.4. Target Gene Prediction and Functional Classification of Drought-Responsive miRNAs*

The target genes of amo-miRNAs were predicted by Target Finder, and 1393 target genes of the 114 miRNAs were predicted, including 545 target genes for the 44 conservative miRNAs and 886 target genes for the 54 *A. mongolicum* specific miRNAs (Table S3). A high proportion of the target genes

were related to cell components production, for example, organelles and cell membrane production. The proportion of the target genes involved in catalytic activities and molecular bindings was also high. Another big portion of the target genes was related to metabolism regulation and stimulus responses. The target genes of the differentially expressed miRNAs were significantly associated with cellular signal transduction and stress responses, including messenger and receptor combination and protein binding of transcription factors as indicated by the GO analysis (Figure 4).

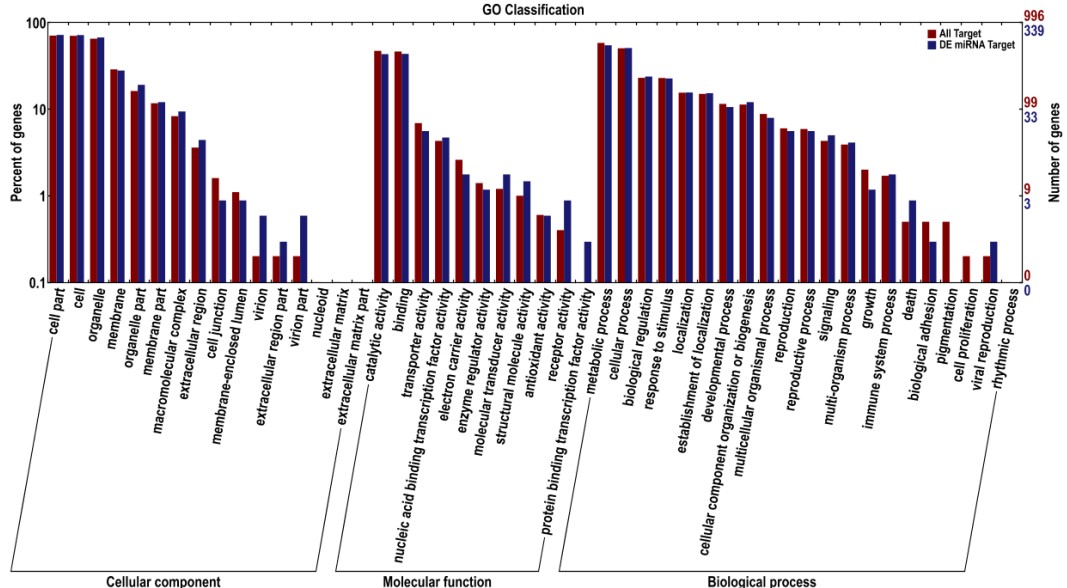

**Figure 4.** Gene Ontology (GO) annotated cluster map of putative target genes of differentially expressed miRNAs in *A. Mongolicum*, classified in three main categories (cellular component, molecular function, and biological process.

A total of 46 KEGG pathways were retrieved from the 153 target genes of the differentially expressed miRNAs. Forty-seven unigenes were identified in these KEGG pathways. Most of the pathways were related to stress resistance, including plant hormone signal transduction, oxidative phosphorylation, inositol phosphate metabolism, plant-pathogen interaction, glycerolipid metabolism, and proteasome and phosphatidylinositol signaling system (Table S4). It was found that ten target genes of several differentially expressed miRNAs (amo-miR102, amo-miR20, amo-miR77, amo-miR65, amo-miR14, amo-miR36, amo-miR37, amo-miR109) were involved in the regulation of plant hormone signal transduction (Figure 5), and hormone signal transduction pathways play an important role in drought tolerance.

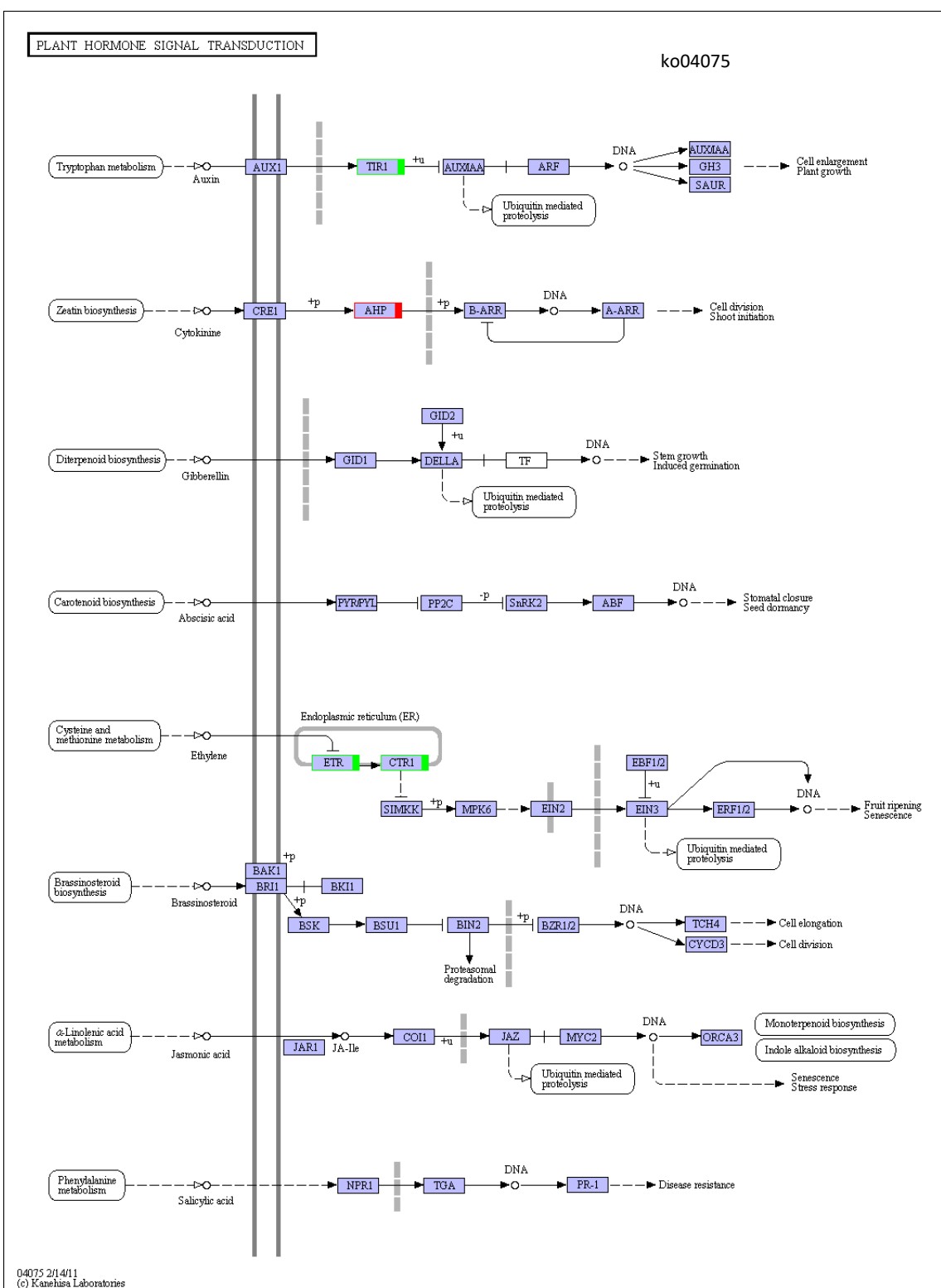

**Figure 5.** Signal transduction of plant hormones. Note: Red was up-regulated and green was down-regulated.

In order to better understand the molecular mechanism of drought tolerance, the target genes of seven differentially expressed drought-responsive miRNAs (amo-miR21, amo-miR62, amo-miR82, amo-miR5, amo-miR77, amo-miR44 and amo-miR17) were predicted via both Target Finder software and psRNATarget database, and the homology were analyzed with MEGA 7.0. Seventeen target

genes of the seven miRNAs were predicted in *A. mongolicum*. Both one-to-one and one-to-many interactions between miRNAs and the target genes were observed. Proteins encoded by these target genes included photosystem II proteins, ubiquitin-proteasome system proteins, PPR (pentatricopeptide repeat) family proteins, and transcription factors such as zinc finger proteins and AP2/ERF and B3 domain-containing proteins. A total of 264 target genes of the seven miRNAs were predicted via psRNATarget in other plant species (Table S5). These miRNA target genes were mainly involved in biosynthesis, transcriptional regulation, material metabolism, signal transduction, stress response, and other biological processes. The target genes predicted in *A. mongolicum* were consistent with those predicted in other plant species, and all of them were related to drought stress response. Interestingly, the target genes of amo-miR17 were not predicted in the psRNATarget database. It might be due to the species-specific and un-conservative nature of miRNAs.

Phylogenetic analysis was done on the target genes of miRNAs in *A. mongolicum* (Figure 6). The 68 target genes of amo-miR77 (Figure 6A) were divided into two big groups and four sub-classes and the genetic relationship among them was relatively close. Unigene30647 and LOC_Os02g50490.1, unigene48598 and GRMZM2G110423_T01, and unigene50222 and AT1G05710.3 and AT1G05710.5 were on the same branch. LOC_Os02g50490.1 had endohydrolysis function; GRMZM2G110423_T01 encoded a zinc finger domain containing protein; and AT1G05710.3 and AT1G05710.5 encoded a basic helix-loop-helix (bHLH) DNA-binding protein. All these target genes were speculated to participate in adverse stress responses, and unigene30647, 48598 and 50222 were likely to be involved in drought stress response. The 30 target genes of amo-miR21 (Figure 6B) were divided into two big groups and the genetic relationship between them was relatively close. Unigene19513 and AT2G02610.1 were on the same branch, and both were related to structural synthesis. The 60 target genes of amo-miR44 (Figure 6C) were divided into two big groups and three sub-classes. Unigene41294 and AT3G19580.2 were on the same branch. The 28 target genes of amo-miR62 (Figure 6D) were divided into two big groups and three sub-classes, and the genetic relationship among them was relatively close. Unigene54336, AT2G25760.2 and AT2G25760.1 were on the same branch; AT2G25760.2 and AT2G25760.1 belonged to the kinase family; unigene54336 encoded a pentatricopeptide repeat-containing protein. All these target genes were speculated to participate in adverse stress response, and unigene54336 may participate in drought stress response. The 29 target genes of amo-miR82 (Figure 6E) were divided into two big groups, and the genetic relationship between them was relatively close. Unigene57301, Traes_6AS_37008F637.2, and GRMZM2G066326_T01 were on the same branch. Traes_6AS_37008F637.2 encoded a NADH-ubiquinone oxidoreductase, and unigene57301 encoded a WD repeat domain-containing protein. All these target genes were speculated to participate in adverse stress response. The 61 target genes of amo-miR5 (Figure 6F) were divided into three big groups, and the genetic relationship among them was relatively close. Unigene49983 and LOC_Os04g16771.1 were on the same branch. Unigene49983 encoded a 47 kda photosystem II protein, and LOC_Os04g16771.1 was a chloroplast chromosome gene.

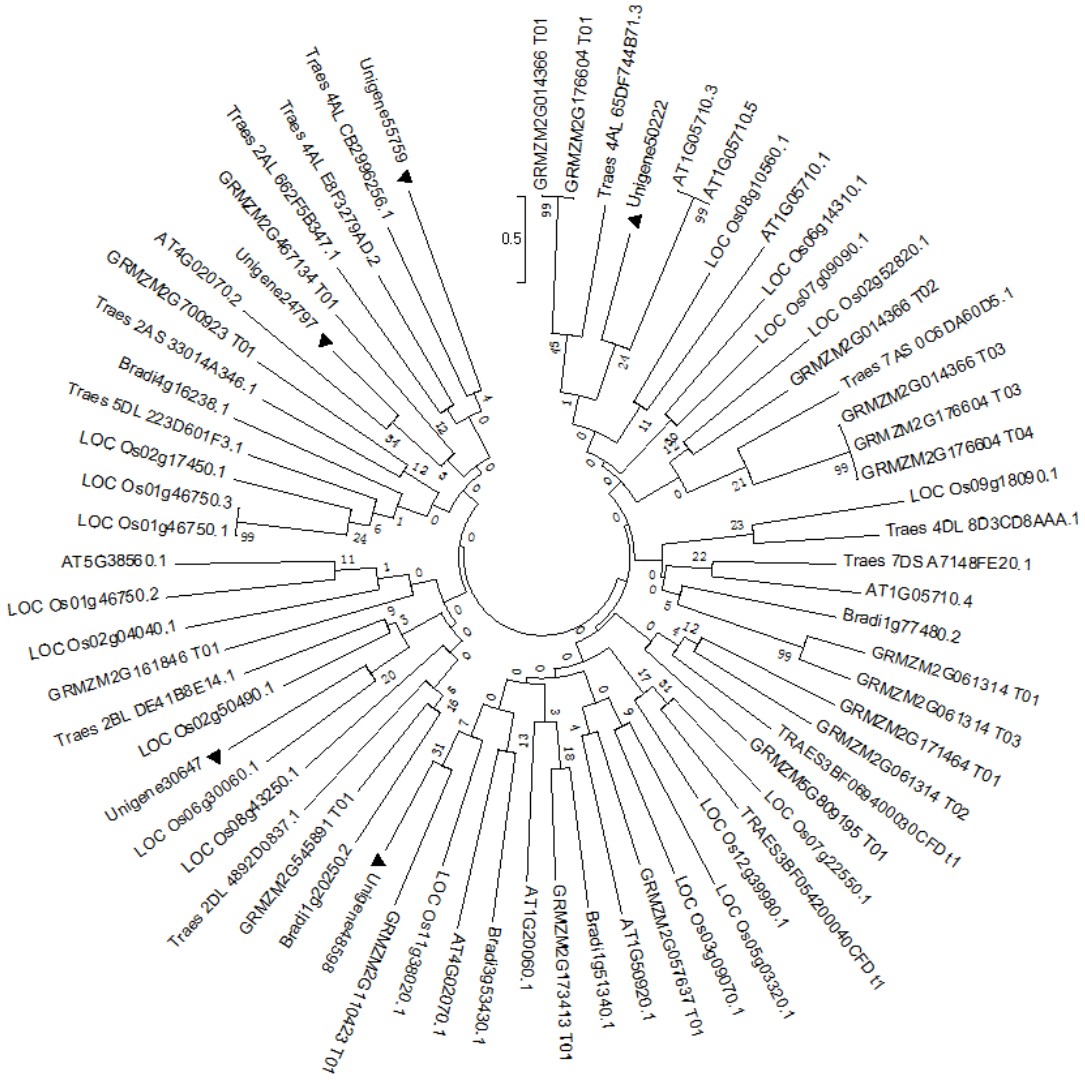

(**A**) Phylogenetic analysis of amo-miR77's target genes.

**Figure 6.** *Cont.*

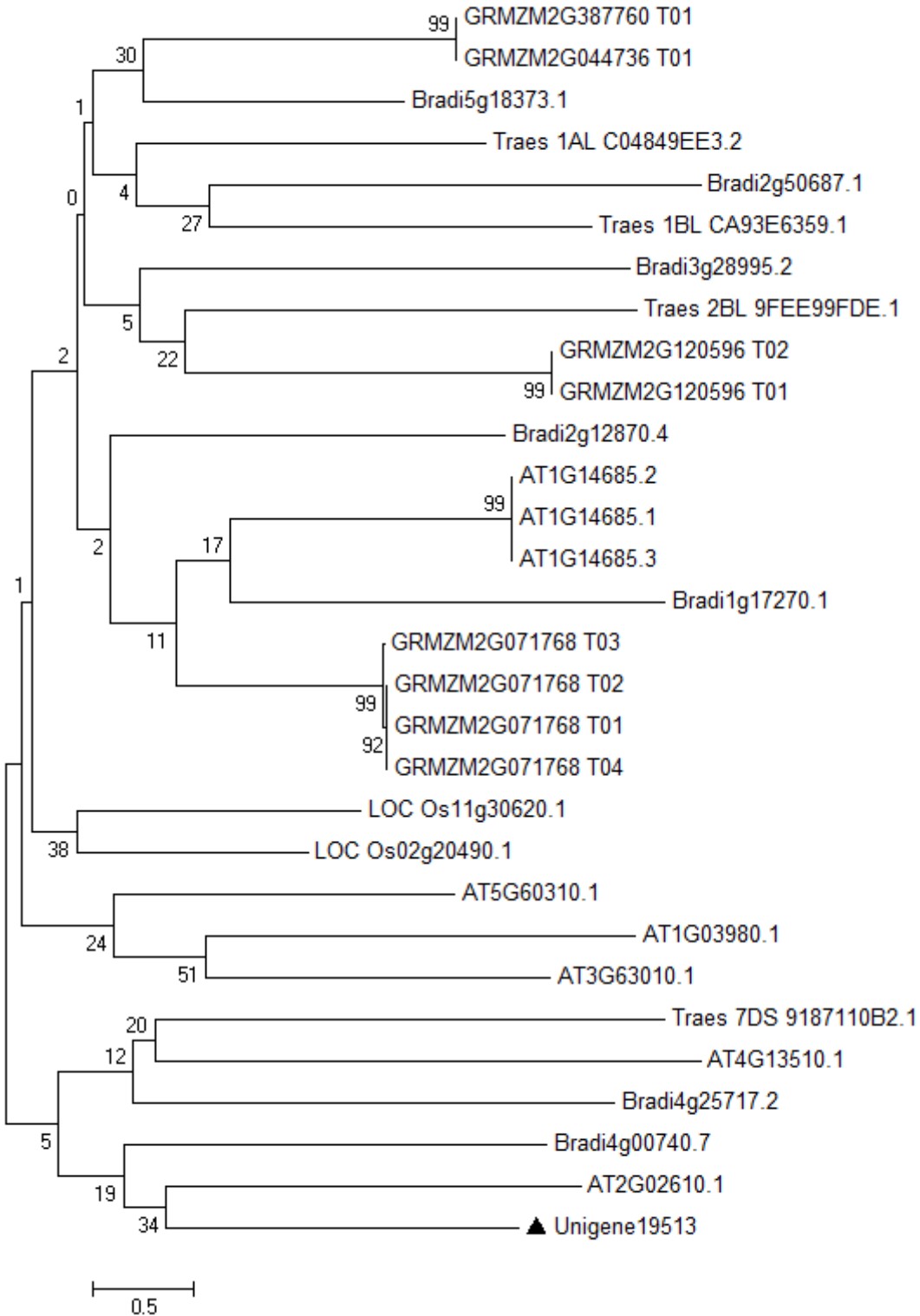

(**B**) Phylogenetic analysis of amo-miR21's target genes.

**Figure 6.** *Cont.*

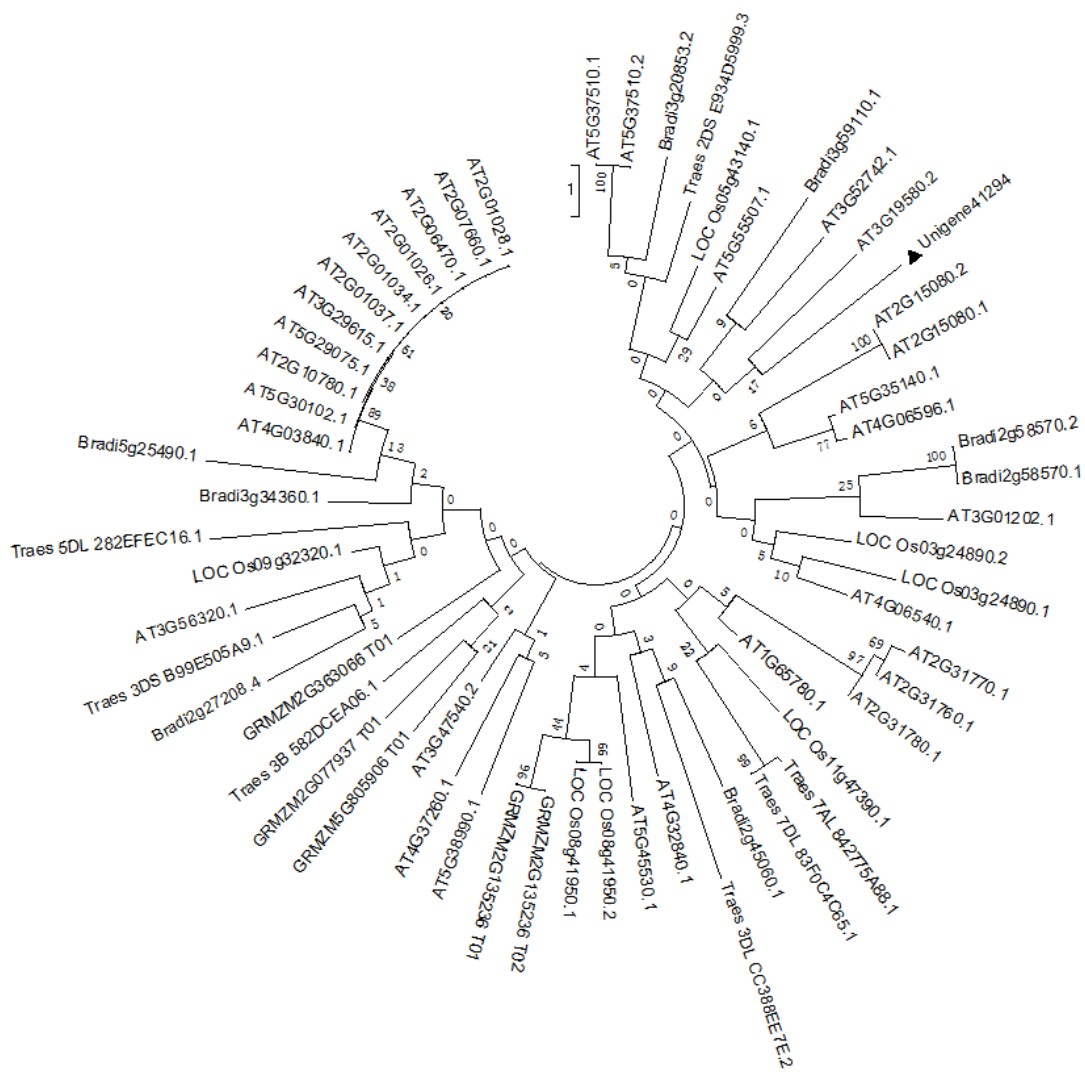

(**C**) Phylogenetic analysis of amo-miR44's target genes.

**Figure 6.** *Cont.*

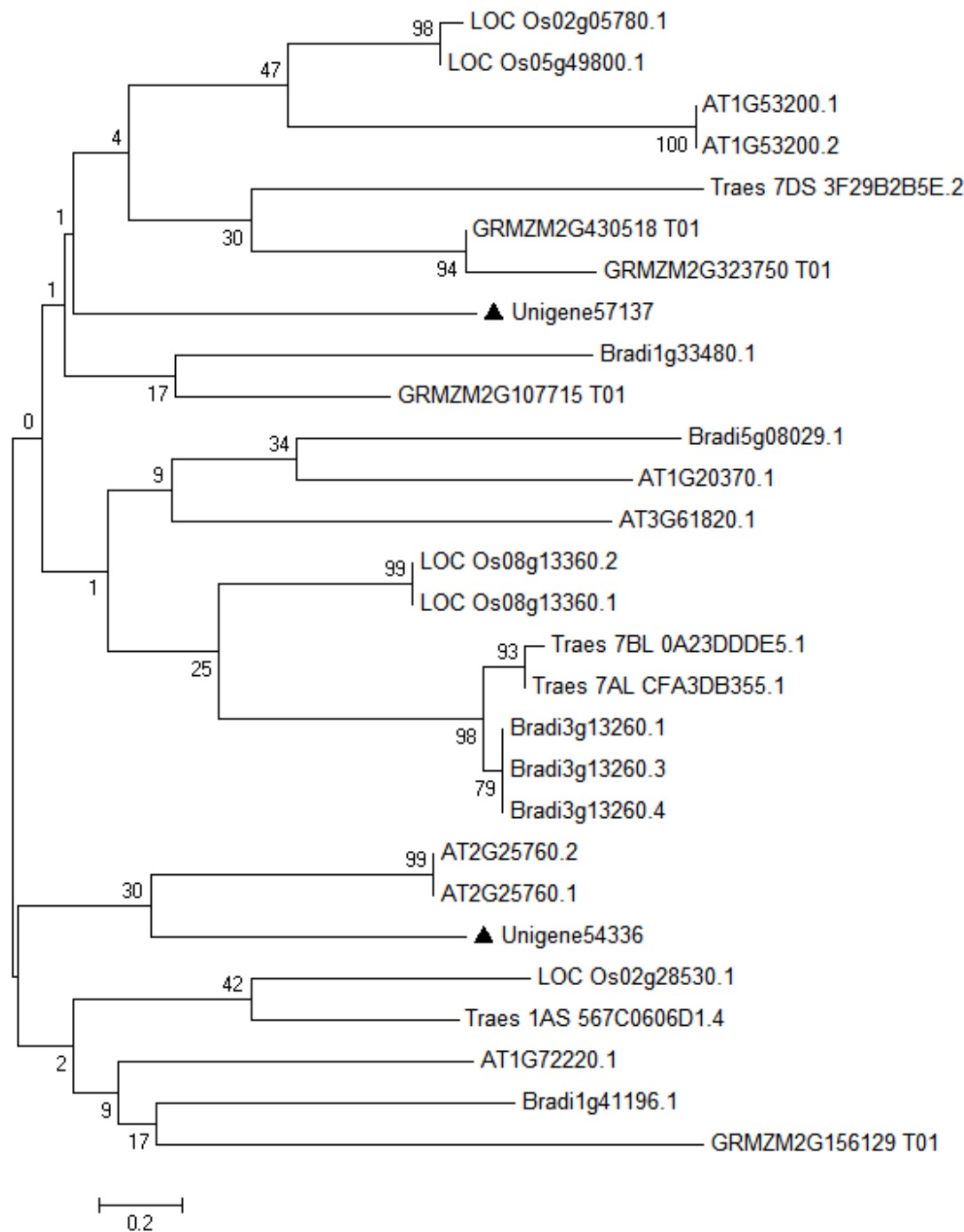

(**D**) Phylogenetic analysis of amo-miR62's target genes.

**Figure 6.** *Cont.*

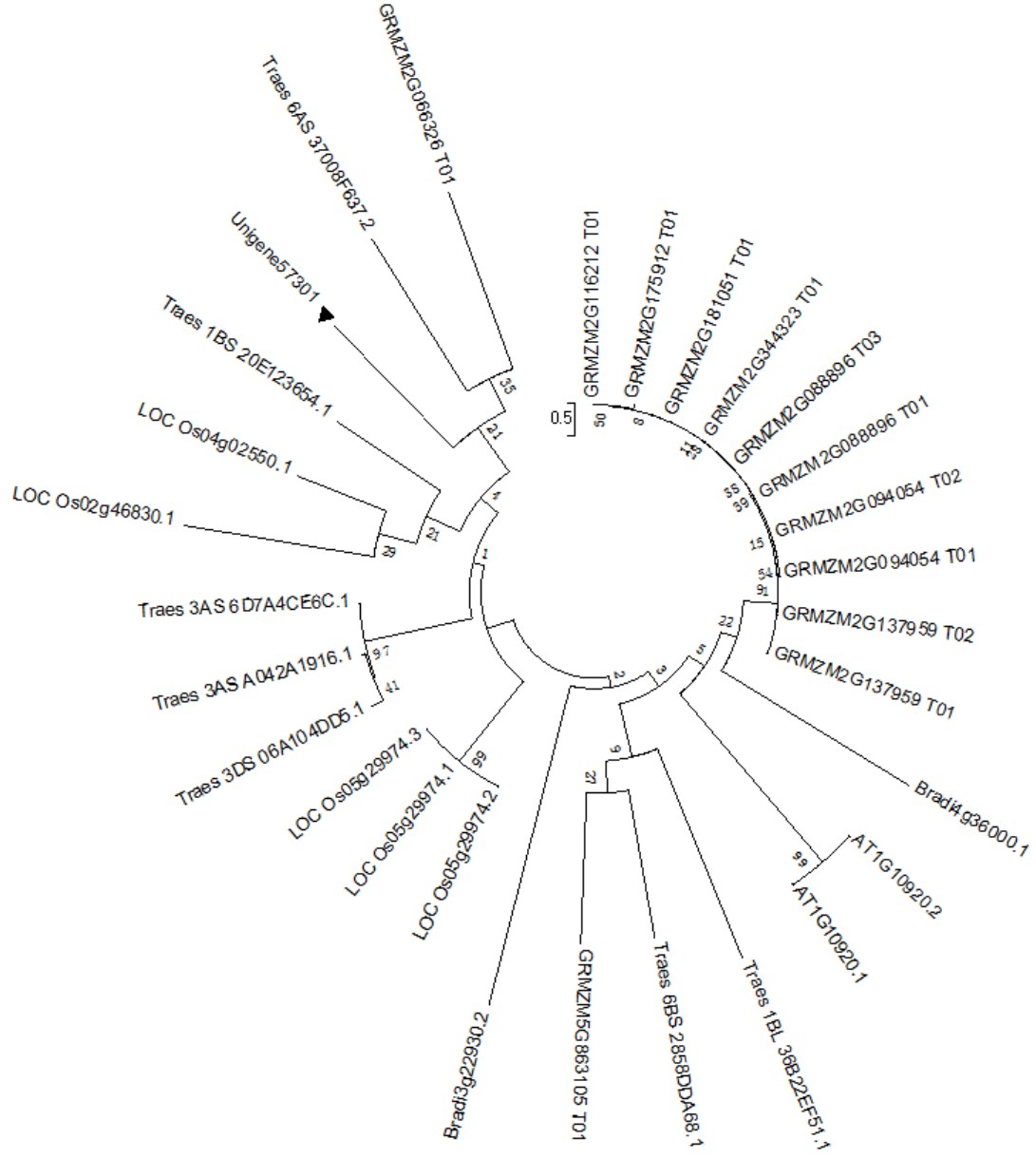

(**E**) Phylogenetic analysis of amo-miR82's target genes.

**Figure 6.** *Cont.*

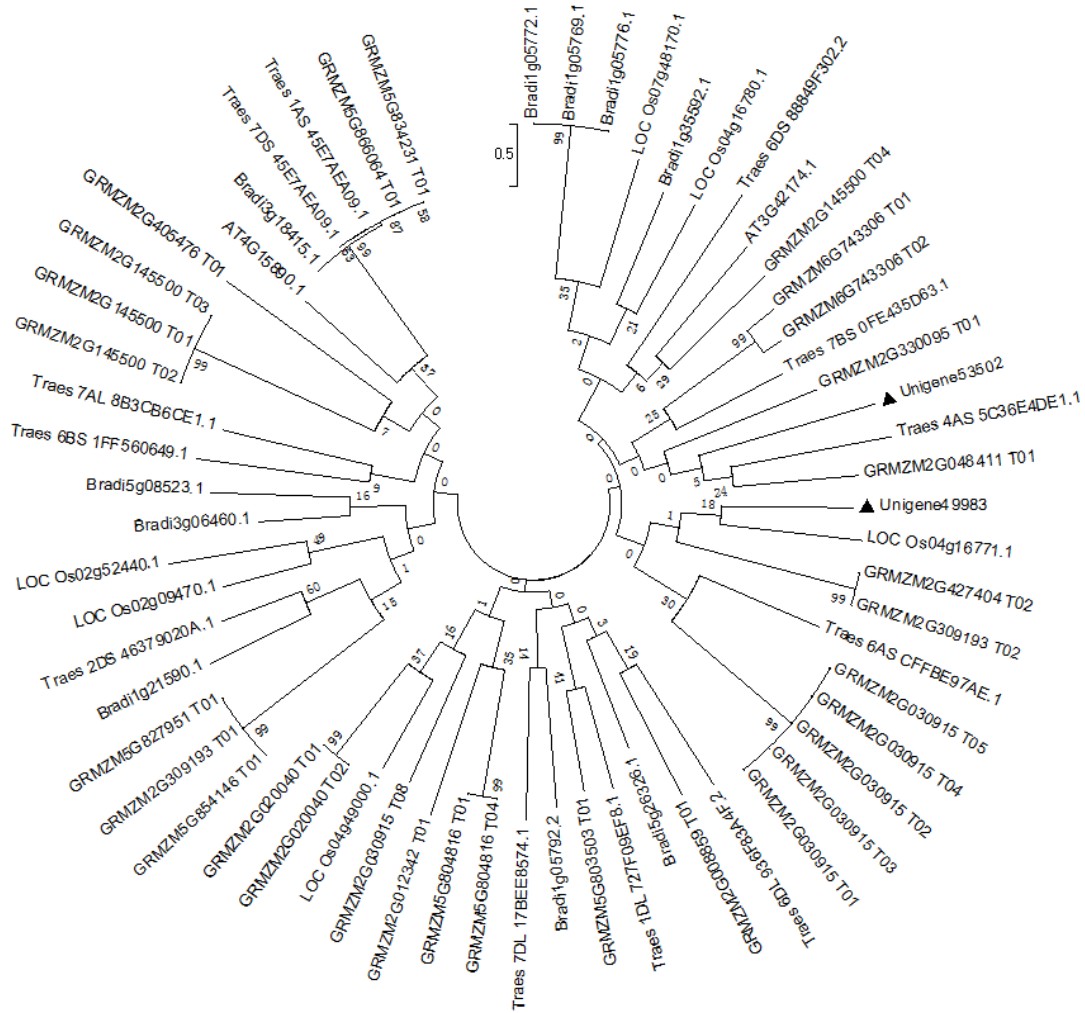

(**F**) Phylogenetic analysis of amo-miR5's target genes.

**Figure 6.** Phylogenetic analysis of 6 miRNAs' target genes in *A. mongolicum*.

### 3.5. Expression Profiles of Seven Drought-Responsive miRNAs in A. mongolicum

The expression of seven drought-responsive miRNAs in *A. mongolicum* were tested by qRT-PCR. The results showed that amo-miR21, amo-miR62, amo-miR82, amo-miR5, and amo-miR77 were significantly up-regulated under drought treatment for 21 d by 2.39-, 2.60-, 2.46-, 3.18-, and 1.54-fold, respectively, compared with the control. For amo-miR21, amo-miR62 and amo-miR82, no significant difference was observed when the *A. mongolicum* plants were under drought treated for 15, 18, or 21 d. It indicated that the expression of these three miRNAs at 15 d post treatment reached the maximum. Interestingly, the expression of amo-miR5 slightly declined at 15 d, and increased sharply at 18 and 21 d. In contrast, the expression level of amo-miR77 increased slowly throughout the drought treatment. Amo-miR44 and amo-miR17 were down-regulated under drought stress. The expression level of amo-miR44 under drought treatment was significantly lower than that of the control. At 21 d post treatment, the expression of amo-miR44 was lowered by 3.23 times. While, the expression level of amo-miR17 decreased rapidly with 11.11 times lower than the control group at 21 d post treatment (Figure 7). In summary, all of the seven newly discovered miRNAs were differentially expressed under drought stress in *A. mongolicum*, which was consistent with the library sequencing data.

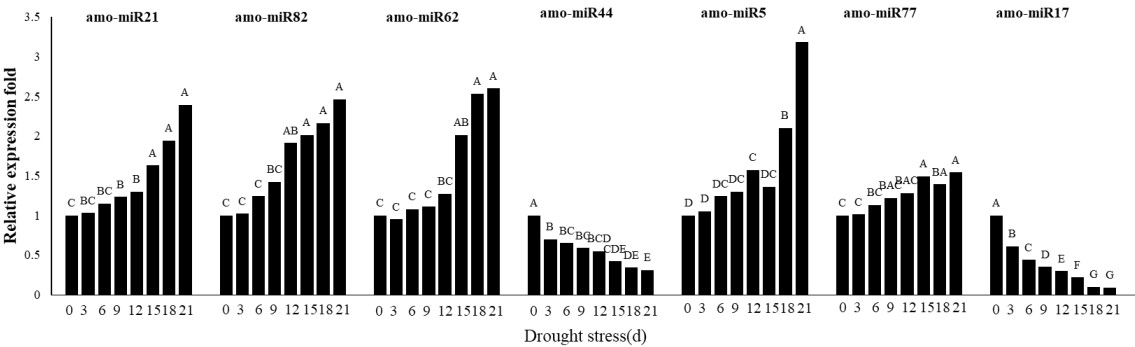

**Figure 7.** Expression profile of miRNAs in *A. mongolicum* under drought stress. Note: When the seedlings of A. mongolicum reached 15 cm high, the plants were divided into two groups randomly, the control group and the drought stressed group. The Sample of drought stressed group were collected at 0, 3, 6, 9, 12, 15,18 and 21 d post drought treatment, and untreated plants were included as the control. The expression levels of seven amo-miRNAs were examined via qRT-PCR. Expression levels were estimated using the $2^{-\Delta\Delta CT}$ method, and U6 was used as a reference gene. Different letters on the column mean significant difference among different treatments at 0.01 level.

### 3.6. *Amo-miR21, Amo-miR5, Amo-miR62 Transgenic* Arabidopsis *Plants Were Tolerant to Drought Stress*

The average seedling rate of the $T_3$ generation of the transgenic *Arabidopsis* plants containing amo-miR21, amo-miR5 and amo-miR62 was more than 60%, indicating that the miRNAs were stable in *Arabidopsis*. Target fragments of 100 bp were amplified from transgenic *Arabidopsis* with specific primers of amo-miR21, amo-miR5, and amo-miR62 by RT-PCR (Figure 8).

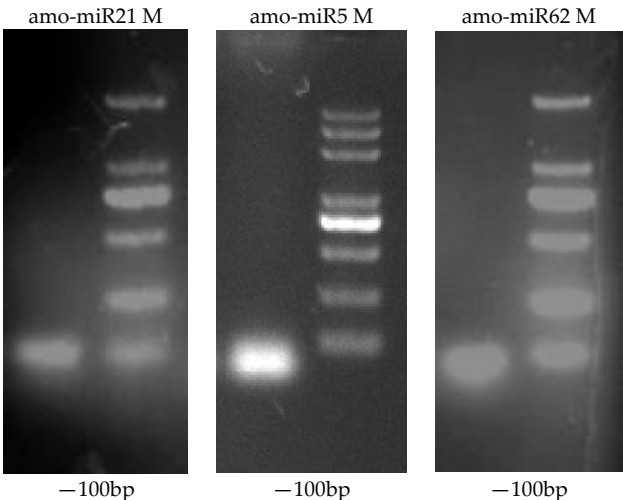

**Figure 8.** RT-PCR amplification of pre-miRNAs in transgenic plants. Note: M: 100 bp of ladder DL2000 (TIANGEN, Beijing, China).

The amo-miR21, amo-miR5, or amo-miR62 transgenic *Arabidopsis* plants were irrigated with 100 mL 15% PEG 6000 solution for drought treatment at flowering period. Data were collected 0, 3, 6, 9, 18, 21 and, 24 h post treatment. Morphological differences were observed between the wild type and the transgenic *Arabidopsis* plants under regular growth condition and 15% PEG-induced drought environment (Figures 9 and 10). The wild-type *Arabidopsis* had more wilting leaves and were more sensitive to 15% PEG-induced drought stress at 18 h post treatment, while the transgenic *Arabidopsis* plants showed fewer wilting leaves. In addition, the transgenic lines displayed significantly higher survival rates, longer roots, and higher root area index compared with the wild type (Figure 11). Amo-miR21, amo-miR5, and amo-miR62 showed the same expression pattern in *Arabidopsis* as that in *A. mongolicum* under drought stress (Figure 12). Taken together, these results indicated that

accumulation of amo-miR21, amo-miR5 and amo-miR62 improved the adaption of *Arabidopsis* to drought stress.

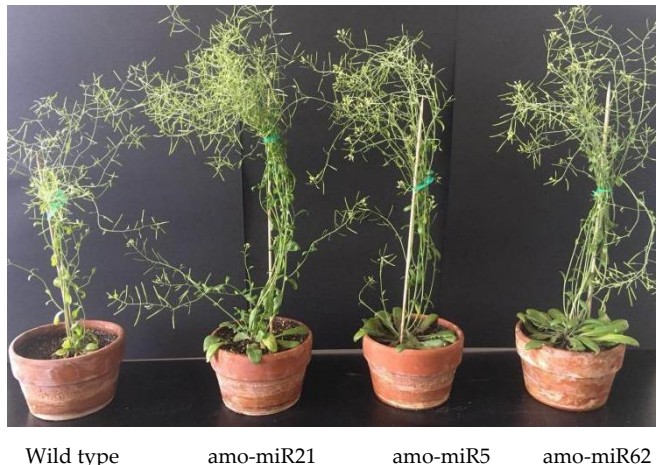

Wild type         amo-miR21        amo-miR5        amo-miR62

**Figure 9.** Phenotype of transgenic *Arabidopsis* lines under normal conditions at flowering stage.

Wild type         amo-miR21        amo-miR5        amo-miR62

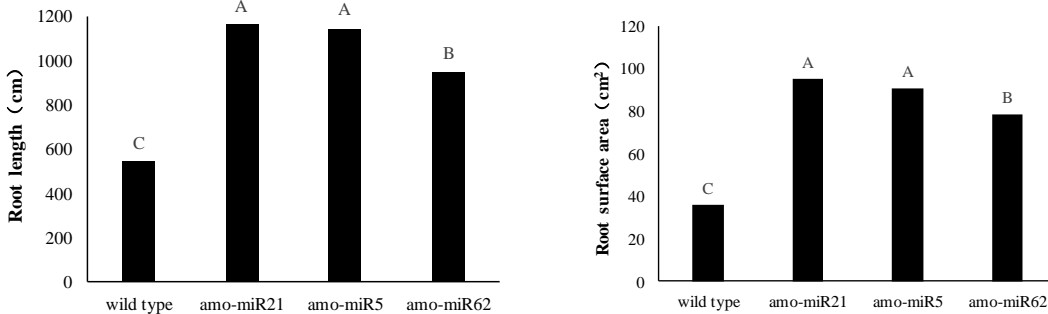

**Figure 10.** Phenotype of transgenic *Arabidopsis* lines at 18 h under 15% PEG treatment.

**Figure 11.** Root characteristics of transgenic *Arabidopsis* lines at 18 h under 15% PEG treatment. Note: Different letters on the column indicate significant difference at 0.01 level.

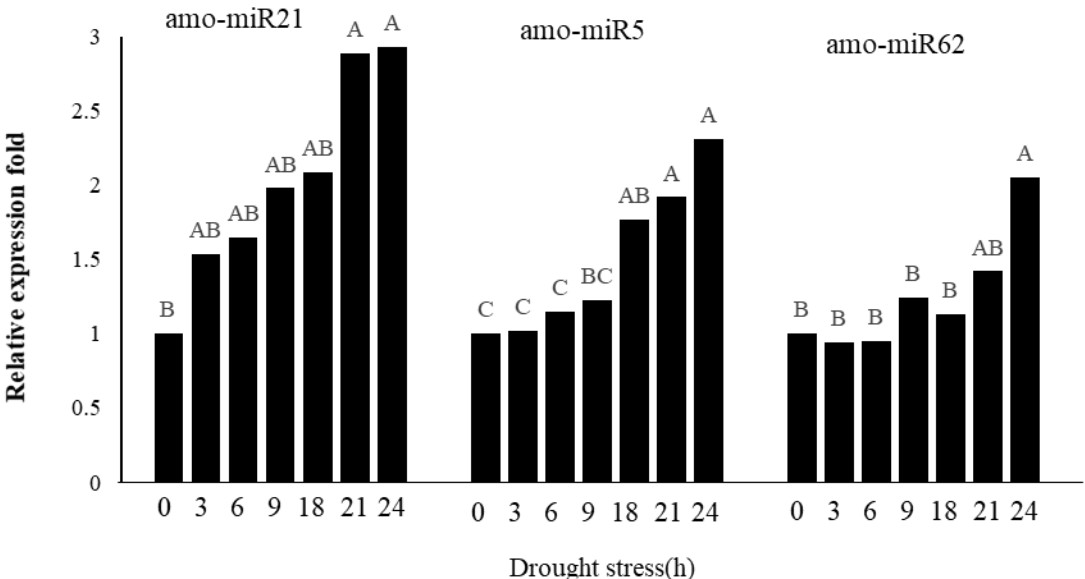

**Figure 12.** Relative expression level of miRNAs in transgenic *Arabidopsis* lines under 15% PEG treatment. Note: Different letters on the column indicate significant difference at 0.01 level.

## 4. Discussion

miRNAs play important roles in plant adaptation to drought stress [20,37]. In recent years, high-throughput sequencing has been widely used to identify miRNAs in plant [19,23,38,39]. *A. mongolicum* is a perennial diploid grass species that has a more complex regulatory mechanism for drought stress. Although some genes involved in drought tolerance, such as *PLD*, *MwLEA*, *MwLEA3*, *MwACT*, *MwGST*, *MWCP*, and *MwLhcb*1 had been studied [40], miRNAs of *A. mongolicum* had not been identified related to drought stress. In this study, amo-miR44 was homologous with ath-miR854a in *A. thaliana*. However, ath-miR854 was up-regulated under drought stress in *A. thaliana* [41], while amo-miR44 was down-regulated in *A. mongolicum* in our study. This may be due to its special processing that the amo-miR44 was generated from the stem rings on the two arms of the precursor. Amo-miR62 was homologous with zma-miR164g-3p of *Z. mays.* The miR164 homologs were a group of plant-specific miRNAs, whose target genes were mainly the NAC transcriptional factor family. The mutants of miR164 in *Arabidopsis* and maize increased the accumulation of its target gene *NACI*, and accordingly promoted the formation of lateral root and increased the drought tolerance in plants; by contrast, over expression of miR164 could decrease the expression of *NACI* and reduce the lateral root formation and plant drought tolerance [42]. Interestingly, conflicting result was also reported. Fang and associates found that in rice, suppressed miR164 adjustment of NAC family genes had a negative regulatory role in rice drought tolerance [43]. In another case, miR164 transgenic *Arabidopsis* plants demonstrated increased root length and germination rate under low temperature to improve plant cold resistance [44]. Similar results were obtained in our study that amo-miR62 was up-regulated in *A. mongolicum* in response to drought stress, and over expression of amo-miR62 in *Arabidopsis* increased the root length and root area index. Thus, the results indicated that homologous miRNAs may function differently in different plant species as post transcriptional regulators in response to adverse stress.

In this study, 17 target genes of seven drought-responsive miRNAs were predicted in *A. mongolicum*, and these target genes have been associated with PPR family proteins, ubiquitin-proteasome system proteins, and transcription factors. The pentatricopeptide repeat (PPR) family proteins are one of the most widely distributed proteins in plants and play a wide and important role in plant growth and development [45]. The ubiquitin-proteasome system proteins are important stress responsive proteins which could be induced by a wide range of biotic and abiotic stresses and increase plant adaption [46]. Transcription factors, such as zinc finger proteins and AP2/ERF transcription factors, were reported

to be essential in plant stress responses [47]. These findings have emphasized the importance of *A. mongolicum* miRNAs as post-transcriptional regulators in drought tolerance.

Online software has been widely used to predict target genes of plant miRNAs. For example, 122 conservative miRNAs and 59 new miRNAs in chickpea were identified, and 358 target genes were predicted by psRNATarget [48]. In this study, psRNATarget was used to predict target genes of amo-miRNAs, and a total of 264 target genes were obtained. Potentially, all the target genes predicted in this study could be transformed into *Arabidopsis* for function validation.

In the phylogenetic analysis, all target genes were close genetically and well clustered. Most unigenes were on the same branch with known genes. For instance, unigene30647 and LOC_Os02g50490.1, unigene48598 and GRMZM2G110423_T01, unigene50222 and AT1G05710.3 (AT1G05710.5), unigene19513 and AT2G02610.1, unigene41294 and AT3G19580.2, unigene54336 and AT2G25760.2 (AT2G25760.1), unigene57301 and Traes_6AS_37008F637.2, and unigene49983 and LOC_Os04g16771.1 were on the same branch. GRMZM2G110423 was reported to be a putative zinc finger domain superfamily protein [49], and AT3G19580 was a zinc-finger protein 2 (ZF2). Previous research suggested that *Arabidopsis* zinc-finger protein 2 (AZF2) was a negative regulator of ABA signaling in seeds [50]. AT1G05710 belonged to a basic helix-loop-helix (bHLH) DNA-binding protein superfamily, and the AtbHLH genes constituted one of the largest families of transcription factors in *A. thaliana*, which had a range of different roles in plant cell and tissue development as well as plant metabolism [51]. AT2G25760 belonged to a protein kinase family, which regulated various physiological processes [52]. Traes_6AS_37008F637 in uniport database was annotated as NADH-ubiquinone oxidoreductase. This indicated that unigene48598, unigene50222, unigene54336 and unigene57301 were most likely to be involved in drought stress response, which could be the focus of further research.

## 5. Conclusions

The work contributes towards to identify the 114 new miRNAs in *A. mongolicum* and understand the molecular function of seven drought related miRNA from *A. mongolicum*. It also provides the basis for new research assumptions.

**Supplementary Materials:** The following are available online at http://www.mdpi.com/2073-4395/9/10/661/s1, Table S1: Summary of unconservative miRNAs in *A. mongolicum*, Table S2: Summary of conservative miRNAs in *A. mongolicum*, Table S3: The target genes of 114 miRNAs in *A. mongolicum*,: Table S4: Integrated function.annotation and KEGG pathways of target genes in *A. mongolicum*, Table S5: 264 target genes of seven miRNAs in *A. mongolicum*.

**Author Contributions:** Y.M. conceived and designed the research. X.Z., B.F., and Y.M. performed the experiments. Y.Z. provided the seeds. Y.M., X.Z., B.F., Z.Y., L.N., X.Y., F.S., and X.L. analyzed the data and wrote the manuscript. All of the authors read and approved the final manuscript.

**Funding:** This work was supported by grants from the National Natural Science Foundation of China (31360573 and 31860670).

**Acknowledgments:** We wish to thank. Y.C. assistance with revising the manuscript.

**Conflicts of Interest:** The authors declare that the research was conducted in the absence of any financial and non-financial relationships that could be construed as a potential conflict of interest.

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
