# Peer review of "Functional Analysis of Three miRNAs in Agropyron mongolicum Keng under Drought Stress"

_agronomy, doi:10.3390/agronomy9100661_

Round 1

Reviewer 1 Report

On the whole, the work is well written, apart from a few minor corrections necessary to improve the manuscript. All observations are given in the attached pdf file in the form of comments. A check of the references, both in the text and in the list, is necessary.

Author Response

Thank you very much for giving us an opportunity to revise our manuscript entitled "Functional analysis of three miRNAs in Agropyron mongolicum Keng under drought stress" I greatly appreciate both your help and that of the referees concerning improvement to this paper. Based on these comments and suggestions, we have made careful modifications on the original manuscript. All changes made and red color highlighted by using the track changes mode in MS Word. I hope that the revised manuscript is now suitable for publication. Below you will find our point-by-point responses to your and the reviewers’ comments.

Reviewer #1:

Comment:Maybe genus Agropyron, tribe Triticeae?

Response: I am sorry to misspelling, which has been corrected to “genus Agropyron, tribe Triticeae”(Line 13).

Comment:Write 17 in full (Seventeen) and not as a number because it is at the begin of the sentence.

Response: According to your suggestion, we have replaced 17 for Seventeen (Line 22).

Comment:Please, re-write from line 38 to line 44 because some concepts has been repeated, indicating also a different reference (1, 2 and 6).

Response: According to your suggestion,I have rewrite from 38 to 47,and have check the references (Line 38 to 47).

Comment:wheat (Triticum aestivum) pathogens

Response: According to your suggestion, we have marked “ (Triticum aestivum L.)” in the correct place (Line 40).

Comment:Please, substitute "from" with "of"

Response: According to your suggestion, we have replaced “from” for “of”.(Line 56).

Comment: 11 is not a Reference in the Reference List. Please, check all references and the correspondence between citations in the text and the references list.

Response: No. 11 is DOI of No.10 Voinnet, O. Origin, biogenesis, and activity of plant microRNAs. J.Cell. 136,669-687; DOI:10.1016/j.cell.2009.01.046.

The initial submission reference is correct, and have a mistake in typesetting. Then I have check all references and the correspondence between citations in the text and the references list.

Comment:Substitute "example" with "case"

Response: According to your suggestion, we have replaced “example” for “case”.(Line 66).

Comment:Eliminate the space before the dot.Write 53 in full and not as a umber because it is at the begin of the sentence

Response: According to your suggestion, we have eliminated the space, and substitute "53" with “Fifty-three”(Line 73).

Comment:For not repeating, please, write "they" instead of "the plants"

Response: According to your suggestion, we substitute "the plants" with “they”(Line 105).

Comment:Please, explain better. After 21 days were the plants watered and than stressed again for 21 days? And again? And, at the same time, were the other group of plants watered continuosely? It is not clear

Response: When the seedlings reached 15 cm high, the plants were divided into two groups randomly, the control group and the drought stressed group. The control group was irrigated daily as described previously, and the drought-stressed plants were not watered for 21days (d). Each treatment was repeated three times. Leaves were sampled at 3d, 6d, 9d, 12d, 15d, 18d, 21d post treatment from both groups. See the methods.

Because of 21 d after drought treatment in Agropyron mongolicum Keng, its’ blade is dry, and only a few plants survival after water again, then confirmed that 21 d is the critical point of water stress.And, at the same time, the control group were the other group of plants watered continuosely to Keep the stable relative water content of soil.

Comment:For completeness, in this point and in the corresponding figures and tables I would also include the meaning of the acronyms.Non-redundant (NR), Gene Ontology (GO), Cluster of Orthologous Groups (COG), Kyoto Encyclopedia of Genes and Genomes (KEGG)

Response: According to your suggestion, we have modified the abbreviation (Line 134 to 236).

Comment:Please, correct the format

Response: I am sorry for wrong format, which has been modified (Line 138)

Comment:here and elsewhere, º of grade often seems to be underlined. Please, correct

Response: We have revised all the ℃(Line 179 to 192)

Comment:Add a space after "primers"

Response: We have add a space after "primers"(Line 166)

Comment:Eliminate the space after μL

Response: We think the space shouldn’t be eliminate between μL and ddH2O, so we didn’t cut out the space(Line 167)

Comment:Add a space before 1.

Response: We have add a space before 1(Line 189).

Comment:LB agar plates containing Kan

Response: According to your suggestion, we have modified LB agar plates containing Kan”(Line 195)

Comment:Correct as [32]

Response: According to your suggestion, we have Correct as [32](Line 202)

Comment:Please, correct the format

Response: According to your suggestion, we have modified format of A. mongolicum(Line 234 to 235)

Comment:The explanation of the acronyms MFE and MFEI has to be reported also in the heading of Table 4.

Response: We think MFE and MFEI are general terms, and it was explained when it’s first appear,then its better to use the acronyms MFE and MFEI in the heading of Table 4(Line 261)

Comment:Add "As showed in Table 4" at the beginning or "(Table 4)" at the end of the sentence.

Response: According to your suggestion, we have add ”(Table 4)" at the end of the sentence(Line 256).

Comment:Please, correct the format so that the close parenthesis do not stand alone

Response: The format has been correct(Line 261).

Comment:In the text, you say "All precursors of the miRNAs were able to form stem loop structures (Figure 2)", but the figure 2 shows only 7 of these. Why? Morover, this figure needs to be fixed because the names of the mi-RNAs are not legible. Response:Because the function of 7 miRNAs were studed, we shows only 7 of these in the figure 2. If necessary, we will add stem loop structures of other miRNAs. This figurehave been fixed correctly.(Line 263 to 269) Comment:Add "As showed in Table 4" at the beginning or "(Table 4)" at the end of the sentence.

Response: According to your suggestion, we have add “(Table 4)" at the end of the sentence(Line 256).

Comment:I would complete the sentence by adding "as indicated by the GO analysis"

Response: According to your suggestion, we have add “as indicated by the GO analysis at the end of the sentence(Line 283).

Comment:Write 17 in full (Seventeen) and not as a number because it is at the begin of the sentence.

Response: According to your suggestion, we have replaced 17 for Seventeen(Line 297).

Comment:Add "(Figure 6A);Add "(Figure 6B);Add "(Figure 6C);Add "(Figure 6D);Add "(Figure 6E);Add "(Figure 6F). Response:According to your suggestion, we have add Figure 6A-6E respectively(Line 310, Line 318,Line 320,Line 322,Line 327,Line 332). Comment:I think it is 48598 and not 48596;I think it is 30647 and not 30467;Please, correct as above;Correct as above;

Response: I am sorry for wrong, and we have modified (Line 311, Line 312,Line 316,Line 317)

Comment:The figure 4 must be improved. The quality is not high and does not seem to be in focus.

Response: According to your suggestion, we have changed it to a clear picture (Line 336).

Comment:Gene Ontology (GO).

Response: According to your suggestion, we have replace “GO” for “Gene Ontology (GO)” (Line 337).

Comment:I think that information about this figure has to be completed by adding, after "Mongolicum" the following text: ", classified in three main categories (cellular component, molecular function and biological process)."

Response: According to your suggestion, we have add “, classified in three main categories (cellular component, molecular function and biological process)” (Line 338 to 339).

Comment:The figure 5 must be improved. The text is not legible.

Response: According to your suggestion, we have changed it to a clear picture (Line 340).

Comment:Add a space after :

Response: According to your suggestion, we have add a space (Line 341).

Comment:What you mean is not clear, and it is different from what is written in the section 2.6. qRT-PCR analysis of Methods.When were the samples collected? In Methods you say: "samples harvested 3d, 6d, 9d, 12d, 15d, 18d, 21d post drought treatment...."

Response: I am sorry for confused expression.When the seedlings of A. mongolicum reached 15 cm high, the plants were divided into two groups random(same to methods).  The growing time is about one month. Now I know the statement is not clear, I have modified it (Line 374 to 378).

Comment:Where are the error bars? They are missing in the figure 7.Please, add "Different letters on the column indicate significant difference at ....."

Response: I am sorry for wrong expression. And I have modified “Different letters on the column mean significant difference among different treatments at 0.01 level.” (Line 380 to381).

Comment:A "T" are missing

Response: I am sorry for wrong. And I have add the missing T (Line 383).

Comment:Please, substitute "100 bp target fragmentes" with "Target fragments of 100 bp" so that the number is not at the beginning of the sentence

Response: According to your suggestion, we have substitute "100 bp target fragmentes" with "Target fragments of 100 bp" (Line 385).

Comment:In the first case the the plants were subjected to stress up to 21 days; in the case of Arabidopsis for 24 hours. How to explain the same pattern?  

Response: The amo-miR21, amo-miR5 or amo-miR62 transgenic Arabidopsis plants were irrigated with 100mL 15% PEG 6000 solution for drought treatment at flowering period. Because the drought resistance is different between Arabidopsis and A. mongolicum, and that the treatment of 15% PEG 6000 is more intense than the treatment of no water. So we selected 24 hours in the case of Arabidopsis.

Comment:Please, move the picture to the left.

Response: I am sorry for wrong typesetting. And I have modified (Line 400).

Comment:Please, add "M: 100 bp DNA Ladder (Company)".

Response: According to your suggestion, we have add”Note: M: 100bp of ladder DL2000 (TIANGEN, Beijing, China )” (Line 401 to 402).

Comment:Please, specify at what time the pictures were taken.

Response: According to your suggestion, we have add “at flowering stage” (Line 405).

Comment:Please, specify miR... in the three figures.

Response: I am sorry for wrong typesetting. And I have modified (Line 408).

Comment:Error bars are missing.

Response: Because the different letters on the column indicate significant difference at 0.01 level, so there is no error bars in two figures (Line 412 and Line 417).

Comment:Please, correct as above.

Response: I am sorry for wrong, and we have modified (Line 459, Line 460,Line 471)

Comment: 52 is the last citation in the text. On the other hands, No. 53-57 are in the references list. Again, please, check the correspondence between citations and final references list.

Response: The initial submission reference is correct, and have a mistake in typesetting. Then I have check all references and the correspondence between citations in the text and the references list.

Thanks again!

Best Regards,

Yanhong MA

Reviewer 2 Report

Please find my Major comments: 

I recommend for an intense correction in the discussion. I don't feel like authors discussed the results rather copy-pasted the results from the results section. Line 35-36: The sentence should be re-written. Re-write the sentence so that it won't lose the meaning. Line 38: Add the scientific name after wheat and before pathogens.  Line 139: What about the negative expression limit? is that -2?  Line 242: What was the min MFE used?  Line 243: Should it be -32?  Temperature is an important factor while generating a stem-loop structure. What was the min temperature used here for drawing the secondary structure? figure 2 is not clear Line 367: it should be the not he.  Line 437-441: This paragraph should be included in the material and method section or in the Result section. 

Author Response

Thank you very much for giving us an opportunity to revise our manuscript entitled "Functional analysis of three miRNAs in Agropyron mongolicum Keng under drought stress" I greatly appreciate both your help and that of the referees concerning improvement to this paper. Based on these comments and suggestions, we have made careful modifications on the original manuscript. All changes made and red color highlighted by using the track changes mode in MS Word. I hope that the revised manuscript is now suitable for publication. Below you will find our point-by-point responses to your and the reviewers’ comments.

Reviewer #2:

Comment:add the scientific name after wheat and before pathogens.

Response: According to your suggestion, we have marked “ (Triticum aestivum L.)” in the correct place (Line 39).

Comment:The sentence should be re-written.Re-write the sentence so that it won't loose the meaning. line (35-36).

Response: According to your suggestion,I have rewrite from 38 to 47,and have check the references (Line 38 to 47).

Comment:what about the negative expression limit?is that -2?

Response: Let's say there are 50 read counts in control and 100 read counts in treatment for gene A. This means gene A is expressing twice in treatment as compared to control (100 divided by 50 =2) or fold change is 2. This works well for over expressed genes as the number directly corresponds to how many times a gene is overexpressed. But when it is other way round (i.e, treatment 50, control 100), the value of fold change will be 0.5 (all underexpressed genes will have values between 0 to 1, while overexpressed genes will have values from 1 to infinity). To make this leveled, we use log2 for expressing the fold change. I.e, log2 of 2 is 1 and log2 of 0.5 is -1.

Comment:what was the min MFE used?

Response: MFE used to represent the combination of the miRNA and its target mRNA level, the greater the absolute value of negative, is combined with the closer, the greater the stability of the structure.

Comment:should it be -32?

Response: I am sorry for wrong. This have been modified -32.50(Line 250).

Comment:Temperature is an important factor while generating the stem loop structure. What was th min temperature used here?

Response: MiRNAs exist a variety of forms,the length of primary transcripts of miRNAs is about 300 ~ 1000 bases;Then generate pri-miRNA, which is then subsequently cut by nuclear RNase III Drosha into 70 ~ 90 nucleotides precursor stem-loop structures (pre-miRNAs), which are processed by the ribonuclease DICER-like 1 (DCL1) into miRNA/miRNA* duplexes.

Comment:figure 2 is not clear.

Response: I am sorry for wrong typesetting. This figure have been fixed correctly.(Line 263 to 269)

Comment:it should be the not he.

Response: I am sorry for wrong. And I have add the missing T (Line 383).

Comment:This paragraph should be included in the material and method section or in Result section.

Response: The purpose of adding this section to the discussion is to explore the reliability of different online software, and this do not repeat with methods.

Comment:Already metioned in the results.

Response: This paragraph is a further discussion to the results,and aimed to indicate the unigenes that were most likely to be involved in drought stress response.

Thanks again!

Best Regards,

Yanhong MA

Round 2

Reviewer 2 Report

I am satisfied with the correction.